# A risk stratification model for high-flow nasal cannula use in patients with coronavirus disease 2019 in Japan: A single-center retrospective observational cohort study

**Ibuki Kurihara** *, **Hitoshi Sugawara**

Division of General Medicine, Department of Comprehensive Medicine 1, Saitama Medical Center, Jichi Medical University, Saitama City, Saitama, Japan

* kibui.rahariku@gmail.com

## Abstract

### Background

The coronavirus disease 2019 (COVID-19) pandemic has put a strain on the healthcare system, and sudden changes in disease status during home treatment have become a serious issue. Therefore, prediction of disease severity and allocation of sufficient medical resources, including high-flow nasal cannula (HFNC), to patients in need are important. We aimed to determine risk factors for the need of HFNC use in COVID-19.

### Methods

This was a single-center retrospective observational cohort study including all eligible hospitalized adult patients aged $\geq$18 years diagnosed with COVID-19 between April 14, 2020 and August 5, 2021 who were treated in the study hospital. The primary outcome is the need for HFNC. Nineteen potential predictive variables, including patient characteristics at hospital admission, were screened using least absolute shrinkage and selection operator and logistic regression to construct a predictive risk score. Accuracy of the risk score was determined using area under the receiver operating characteristic curve.

### Results

The study cohort included 148 patients. The rate of the need for HFNC was 22.9%. Among the 19 potential variables, percutaneous oxygen saturation ($SpO_2$) <92% (odds ratio [OR] 7.50, 95% confidence interval [CI] 2.806–20.82) and IL-6 (OR 1.021, 95% CI 1.010–1.033) were included in developing the risk score, which was termed interleukin (IL)-6-based COVID-19 severity (IBC-S) score.

### Conclusions

The IBC-S score, an easy-to-use risk score based on parameters available at the time of hospital admission, predicted the need for HFNC in patients with COVID-19. The IBC-S

**Data Availability Statement:** All relevant data are within the paper and its Supporting Information files.

**Funding:** The authors received no specific funding for this work.

**Competing interests:** The authors have declared that no competing interests exist.

**Abbreviations:** COVID-19, coronavirus disease 2019; HFNC, high-flow nasal cannula; $SpO_2$, percutaneous oxygen saturation.

score based on interleukin-6 and $SpO_2$ might aid in determining patients who should be transported to a tertiary medical institution or an isolation facility.

## Introduction

The coronavirus disease 2019 (COVID-19) pandemic has claimed approximately 15 million lives as of September 2022 [1]. While approximately 80% of patients with COVID-19 experience a mild disease course and recover, the remaining 20% of the infected patients rapidly progress to severe COVID-19 [2] High-flow nasal cannula (HFNC) can deliver high concentrations of humidified oxygen with low positive end-expiratory pressure and facilitate the elimination of carbon dioxide, thereby rapidly relieving the symptoms of acute hypoxemic respiratory failure [3]. In contrast, HFNC may promote further lung injury (e.g., self-inflicted lung injury) through vigorous breathing efforts [4]. In the COVID-19 pandemic, HFNC use significantly reduced the need for mechanical ventilation support and shortened the time to clinical recovery compared with conventional low-flow oxygen therapy [5]; therefore, HFNC has been played an important role in the management of patients with severe or critical COVID-19 [6].

In Japan, the COVID-19 pandemic has put a strain on the healthcare system and the sudden change in disease status during home treatment has become a serious problem [7]. Therefore, predicting the severity of COVID-19 and allocating sufficient medical resources, including HFNC, to patients in need are important. We aimed to determine the risk factors for HFNC use in patients with COVID-19 by examining the medical records of patients treated in our medical center during the COVID-19 Alpha, Beta, and Delta surges in Japan.

## Materials and methods

### Study design, participants, and setting

This was a single-center retrospective observational cohort study including all adult patients aged ≥18 years that were diagnosed with COVID-19 and hospitalized in the Division of General Medicine, Saitama Medical Center, Jichi Medical University between April 14, 2020 and August 5, 2021. The study period covered the COVID-19 Alpha, Beta, and Delta surges in Japan.

### Primary outcome, data collection, and processing

The primary outcome is the need for HFNC. HFNC was administered to patients who were unable to maintain an $SpO_2$ level of above 92% and respiratory rate below 25 breaths per minute, even when they received standard oxygen administered through a face mask at the flow rate of 10 L/min or higher [8, 9]. In all patients, the COVID-19 diagnosis was confirmed based on a positive nucleic acid amplification test using oropharyngeal, nasopharyngeal, or oropharyngeal/nasopharyngeal swab samples. We excluded patients who needed oxygenation flow rate ≥10 L/min, needed HFNC and mechanical ventilation, or died on admission day. The study data were collected from electronic medical records between September 15, 2021 and March 31, 2023.

### Sample size estimation

G*Power version 3.1.9.2 (Heinrich Heine University Duesseldorf, Duesseldorf, Germany) was used to determine the required sample size using the following parameters: test family, z test;

statistical test, logistic regression; type of power analysis, a priori: compute required sample size—given α, power, and effect size; tails, two; odds ratio, 8.142 (alternative hypothesis was assumed to be moderately correlated at 0.3, and null hypothesis was assumed to be 0.05); α error probability, 0.05; power ($1 - β$ error probability), 0.8; $R^2$ other X, 0.26 (i.e., the regression equation is accurate.); X distribution, exponential; X parm λ, 0.3. The calculated sample size was 44.

## Potential predictive variables

We referred to the previous studies and extracted potential predictive variables [2, 10, 11]. Potential predictive variables were patient characteristics at the time of hospital admission and included demographic variables (age, sex, and smoking status), medical history (diabetes mellitus, hypertension, coronary artery disease, chronic kidney disease, chronic heart failure, and cancer), physical parameters (body mass index and percutaneous oxygen saturation [$SpO_2$]), laboratory parameters (lymphocyte count, C-reactive protein, albumin, direct bilirubin, lactate dehydrogenase, ferritin, D-dimer, and interleukin 6 [IL-6]), and prehospital treatments (steroid, remdesivir, tocilizumab, and baricitinib).

## Ethics approval

The current study was approved by the Institutional Clinical Review Board of Saitama Medical Center, Jichi Medical University (approval no: Clinical # S21-024). In accordance with the ethical guidelines for medical and health research involving human subjects in Japan, written informed consent was not required due to the retrospective study design. The study was conducted with the online opt-out method accessed through the hospital website.

## Statistical analysis

The 19 potential predictive variables were entered into analysis to construct a predictive risk score for HFNC use. The Mann–Whitney *U* and Pearson's chi-square tests were used to compare continuous and categorical baseline characteristics, respectively. The least absolute shrinkage and selection operator method was used to minimize the potential collinearity of variables from the same patient and the over-fitting of variables. Imputation for missing variables was considered if missing values were lower than 20%. The selected variables and treatment before admission which was a confounder, were included in the multiple logistic regression analysis. The predictive value of risk stratification models was assessed using area under the receiver operating characteristic curve. For continuous variables included in the model, parameter values with the equal sensitivity and specificity and the highest positive predictive value were chosen as cut-off values [12]. After the creation of the risk scoring model using the selected variables, bootstrapped logistic regression was used to verify internal validity. All statistical tests were two-tailed with significance set at a P value of <0.05.

The Stata/SE 16.2 software (StataCorp, College Station, TX, USA) was used for all statistical analyses.

## Results

### Baseline information of the participants

The study flow chart is shown in Fig 1. During the study period, a total of 152 patients were admitted with COVID-19 in the study institution. The actual study cohort size was markedly above the estimated sample size (152 versus 44), which increased the power of the study. Of these 152 patients, 4 patients who needed oxygenation flow rate ≥10 L/min, needed HFNC

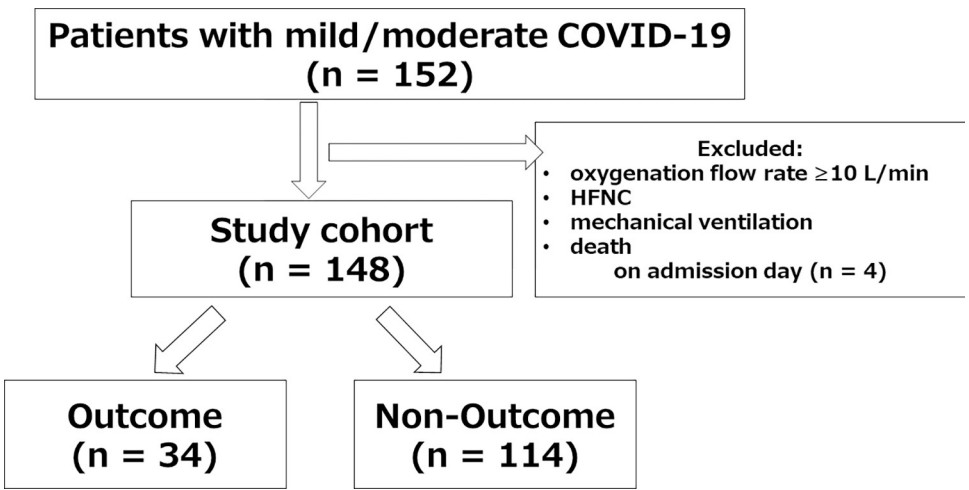

**Fig 1. Study flow diagram.** Outcome indicates requirement of HFNC, Non-Outcome indicates no requirement of HFNC. HFNC: high-flow nasal cannula; COVID-19: coronavirus disease 2019.

and mechanical ventilation, or died on admission day were excluded; therefore, the final analysis included 148 patients.

Table 1 shows the demographic and clinical characteristics of the cohort. Briefly, the median age (interquartile range) was 58.5 (23–84) years, 43 patients (30%) of the patients were female, and 18.4% of the patients had at least one comorbidity. Of the 148 patients, 34 patients (22.9%) required HFNC, 9 (6.1%) patients required invasive ventilation, and 13 patients (8.8%) died. Additionally, 12 patients (8.1%) were admitted to the intensive care unit after the day of admission.

## Selection of predictors

A total of 19 variables that were measured at the time of admission were included in the least absolute shrinkage and selection operator regression, which identified $SpO_2$ and IL-6 as significant predictors of the need for HFNC. In the logistic regression model including these two parameters, both an $SpO_2$ <92% (odds ratio [OR] 7.50, 95% confidence interval [CI] 2.806–20.82; P < 0.001) and IL-6 (per 10 pg/ml increase, OR 1.23, 95% CI 1.010–1.038; P < 0.001) were independent and significant predictors of the need for HFNC and were included in the risk score (Table 2).

## Construction of the risk score

The cut-off IL-6 value of 19.5 pg/mL exhibited equal sensitivity and specificity with the highest positive predictive value of 63.2 pg/mL; therefore, this value was used to create the IL-6-based COVID-19 severity [IBC-S] score to determine the risk of the need for HFNC (Table 3). The IBC-S risk score ranged from 0 to 3.

## Performance of the risk score

The bootstrap analysis for internal validation indicated that the mean area under the receiver operating characteristic curve based on the cohort data was 0.834 (OR 4.762, 95% CI 2.198–10.31; P < 0.001). An IBC-S score of <1 had a sensitivity of 93.94% and a negative likelihood ratio of 0.13. An IBC-S score of >3 had a specificity of 98.15% and a positive likelihood ratio of 19.63 (Table 4).

**Table 1. Baseline patient characteristics.**

| | Total patients (n = 148) | Outcome* (n = 34) | Non-outcome** (n = 114) | P value |
|---|---|---|---|---|
| Age | 58.5 (23–84) | 60 (44–89) | 57.5 (22–83) | 0.037 |
| Male, n (%) | 105 (70%) | 25 (73.5%) | 80 (70.1%) | 0.705 |
| BMI (kg/m$^2$) | 24.69 (17.9–33.5) | 25.1 (16.8–33.3) | 24.6 (17.9–33.8) | 0.521 |
| Diabetes, n (%) | 51 (34.4%) | 18 (52.9%) | 33 (28.9%) | 0.01 |
| Hypertension, n (%) | 77 (52.0%) | 25 (73%) | 52 (45.6%) | 0.004 |
| Ever smoker, n (%) | 71 (48.3%) | 14 (41.1%) | 57 (50.4%) | 0.343 |
| Coronary artery disease, n (%) | 11 (7.4%) | 4 (11.7%) | 7 (6.1%) | 0.272 |
| Chronic heart failure, n (%) | 6 (4.0%) | 3 (8.8%) | 3 (2.6%) | 0.108 |
| Chronic kidney disease, n (%) | 16 (10.8%) | 4 (11.76%) | 12 (10.5%) | 0.838 |
| SpO$_2$ on room air <92%, n (%) | 70 (47.3%) | 28 (82.3%) | 42 (36.8%) | 0 |
| Malignancy, n (%) | 28 (18.9%) | 6 (17.6%) | 22 (19.3%) | 0.829 |
| Laboratory tests | | | | |
| Albumin (g/dL) | 3.6 (2.5–4.5) | 3.4 (1.9–4) | 3.7 (2.6–4.5) | 0.001 |
| D-bil (mg/dL) | 0.2 (0.1–0.64) | 0.24 (0.12–0.87) | 0.19 (0.1–0.55) | 0.029 |
| LDH (U/L) | 296 (146–541) | 358 (214–591) | 264 (143–541) | 0 |
| CRP (mg/dL) | 3.95 (0.09–20.14) | 7.65 (0.76–29.2) | 2.94 (0.06–14.2) | 0 |
| Lymphocyte(/μL) | 804 (294–1938) | 516 (172–1286) | 891 (392–2172) | 0 |
| Ferritin (ng/mL) | 359 (60–1486) | 467 (139–1769) | 310 (51–1486) | 0.014 |
| D-dimer (μg/mL) | 1.15 (1–4.8) | 1.4 (1–5.1) | 1.1 (1–4.8) | 0.038 |
| IL-6 (pg/mL) | 15 (2.2–139) | 31.7 (5.6–186) | 13.05 (1.3–65.5) | 0 |
| Therapy before-admission | | | | |
| Steroids, n (%) | 25 (16.8%) | 10 (29.41%) | 15 (13.16%) | 0.026 |
| Remdesivir, n (%) | 6 (4.0%) | 4 (3.5%) | 2 (5.88%) | 0.538 |
| Tocilizumab, n (%) | 0 | 0 | 0 | |
| Baricitinib, n (%) | 1 (0.68%) | 1 (0.88%) | 0 | 0.534 |
| Clinical course | | | | |
| The need for HFNC | 34 (22.9%) | 34 (100%) | 0 | |
| The need for mechanical ventilation | 9 (6.1%) | 9 (26.4%) | 0 | |
| death | 12 (8.1%) | 12 (35.2%) | 0 | |

*Indicates requirement of HFNC, and

**indicates no requirement of HFNC.

Continuous data are shown as medians with interquartile ranges.

The Mann–Whitney and Pearson's chi-square tests were used.

BMI, body mass index; SpO$_2$, oxygen saturation; D-bil, direct bilirubin; LDH, lactate dehydrogenase; CRP, C-reactive protein; IL-6, interleukin-6; HFNC, high-flow nasal cannula

**Table 2. Association of SpO$_2$ <92% and IL-6 with the risk of the need for HFNC.**

| | Crude | | Multivariable-adjusted | |
|---|---|---|---|---|
| | OR (95% CI) | P value | OR (95% CI) | P value |
| SpO$_2$ <92% | 8.00 (3.06–20.9) | <0.001 | 7.50 (2.80–20.0) | <0.001 |
| IL-6 (per 10 pg/mL increase) | 1.19 (1.07–1.32) | 0.001 | 1.23 (1.10–1.38) | <0.001 |

Logistic regression models were used to estimate odds ratios, 95% confidence intervals, and P values.

The multivariable logistic regression model included two likely confounders: before-admission therapy with steroids and remdesivir.

SpO$_2$, oxygen saturation; IL-6, interleukin-6; CI, confidence interval; OR, odds ratio

**Table 3. IL-6 based COVID-19 severity score to predict the need for HFNC.**

| Variable | IBC-S score |
|---|---|
| SpO$_2$ on room air (%) | |
| ≥92 | 0 |
| <92 | 1 |
| IL-6 (pg/mL) | |
| <19.5 | 0 |
| 19.5–63.2 | 1 |
| 63.2 | 2 |

IBC-S, interleukin-6-based COVID-19 severity; SpO$_2$, oxygen saturation; IL-6, interleukin-6.

**Table 4. Performance of the IBC-S score.**

| Cut-off value | Sensitivity (%) | Specificity (%) | Positive likelihood ratio | Negative likelihood ratio |
|---|---|---|---|---|
| ≤1 | 93.9 | 43.5 | 1.66 | 0.13 |
| ≥3 | 36.3 | 98.1 | 19.6 | 0.64 |

## Discussion

Early identification of patients with COVID-19 at increased risk for HFNC use is beneficial for the proper use of medical resources. In the present study, we found that the combination of SpO$_2$ and IL-6 at admission may predict critical respiratory illness in patients hospitalized with COVID-19.

Simple risk scoring strategies are important in determining where to care for patients. The IBC-S score provides a simple assessment approach based on a small number of predictor variables determined at the time of admission. The IBC-S score has only two predictor variables; in comparison, the 4C mortality score has eight predictor variables and the COVID-GRAM score has ten predictor variables [2, 13]. Moreover, the IBC-S score requires only physical examination findings and the measurement of IL-6 and does not require medical history, blood gas tests, or imaging studies. In the COVID-19 era, simple risk scores are important since imaging studies and more sophisticated tests may not be available in many situations.

SpO$_2$ is one of the two variables of the IBC-S score. The main pathologic feature of COVID-19 is viral pneumonia with alveolar edema and blockage of small bronchi, leading to the compromise of pulmonary gas exchange and reduced oxygen saturation, which is a major indicator of disease severity [14]. Since arterial blood gas analysis is an invasive and complex test, SpO$_2$ measurement is more frequently used to estimate the arterial oxygen partial pressure in primary care settings [15] A study previously reported SpO$_2$ as one of the predictors of inpatient mortality in patients with COVID-19 [2]. SpO2 is one of the preeminent predictors of severe COVID-19.

High IL-6 levels are a predictor of severe COVID-19. IL-6 is a cytokine in cell signaling and the regulation of immune cells. IL-6 has a strong proinflammatory effect with multiple biologic functions and plays an important role in inflammation [16]. In patients with COVID-19 complicated by acute respiratory distress syndrome, hyperactivation of the immune system with prominent IL-6 response can lead to organ dysfunction [11]. Several systematic reviews and meta-analyses revealed that IL-6 inhibitors such as tocilizumab were associated with reduced mortality in patients with COVID-19 [11, 17]. In one systematic review and meta-analysis [9, 15], IL-6 was one of the predictors of ventilator management and death [11, 17]. IL-6 can be measured at common diagnostic facilities, and there are commercially available IL-6

measurement ELISA kits, enabling measurement in primary care settings. The evaluation of IL-6 and $SpO_2$ may aid in identifying patients who should be transported to an appropriate medical institution or an isolation facility.

The present study has several limitations that should be acknowledged. First, the external validity of the IBC-S score was not tested and future studies using other cohorts are warranted to confirm its validity. Second, the study population did not include patients infected with the Omicron variants, which have been increasing in prevalence worldwide [18]. Several Omicron subvariants exhibit replication advantage over prior variants [19]and can evade human immunity to a greater extent than the prior variants [20]. Additionally, the Omicron subvariants might be associated with less severe disease than the other variants [21]. Ebell et al. reported that a lower $SpO_2$ was associated with critical illness in patients infected with the Omicron variant, although the association of IL-6 with disease severity was not investigated [22]. It is possible that the IBC-S score might show different results in the evaluation of patients infected with the Omicron variant. Finally, the current study cohort did not include patients who were vaccinated. Future studies should consider evaluating the utility of the IBC-S score in other cohorts including those infected with other variants and those who are vaccinated.

## Conclusions

The IBC-S score, an easy-to-use risk score based on parameters that can be obtained at the time of hospital admission, predicted critical respiratory failure. The evaluation of IL-6 and $SpO_2$ might aid in determining patients who should be transported to a tertiary medical institution or an isolation facility.

## Supporting information

**S1 Data.**
(XLSX)

## Acknowledgments

We would like to thank Drs. Takahiko Fukuchi, Hiroshi Hori, and Hanako Yoshihara for their medical care of the study participants. We also thank Enago for providing English proofreading services.

## Author Contributions

**Conceptualization:** Ibuki Kurihara.

**Data curation:** Ibuki Kurihara.

**Formal analysis:** Ibuki Kurihara.

**Investigation:** Ibuki Kurihara.

**Project administration:** Ibuki Kurihara.

**Supervision:** Hitoshi Sugawara.

**Validation:** Ibuki Kurihara, Hitoshi Sugawara.

**Visualization:** Ibuki Kurihara.

**Writing – original draft:** Ibuki Kurihara.

**Writing – review & editing:** Hitoshi Sugawara.

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
