## [Decision Letter · Decision Letter 0]

3 Sep 2023

PONE-D-23-24917A risk stratification model for high-flow nasal cannula use in patients with coronavirus disease 2019 in Japan: a single-center retrospective observational cohort studyPLOS ONE

Dear Dr. Kurihara,

Thank you for submitting your manuscript to PLOS ONE. After careful consideration, we feel that it has merit but does not fully meet PLOS ONE’s publication criteria as it currently stands. Therefore, we invite you to submit a revised version of the manuscript that addresses the points raised during the review process.

ACADEMIC EDITOR: Please also acknowledge that while creating this new model, the authors have not tried to validate this new model externally.

We look forward to receiving your revised manuscript.

Kind regards,

Gilbert Sterling Octavius

Academic Editor

PLOS ONE

Journal Requirements:

Reviewers' comments:

Reviewer's Responses to Questions

**Comments to the Author**

1. Is the manuscript technically sound, and do the data support the conclusions?

Reviewer #1: Yes

Reviewer #2: Partly

2. Has the statistical analysis been performed appropriately and rigorously? 

Reviewer #1: I Don't Know

Reviewer #2: Yes

3. Have the authors made all data underlying the findings in their manuscript fully available?

Reviewer #1: Yes

Reviewer #2: Yes

4. Is the manuscript presented in an intelligible fashion and written in standard English?

Reviewer #1: Yes

Reviewer #2: Yes

5. Review Comments to the Author

Reviewer #1: The authors present their single-center retrospective observational cohort study aiming to identify predictors of respiratory failure. I found this to be a simple and elegant study.

Major comments:

1. The definition of the primary outcome (respiratory failure) covers a wide spectrum of pathology from requiring more than 10L/ min oxygen to requiring mechanical intubation or death. I don't find this to be clinically useful. Is it possible for the authors to look at more meaningful clinical categories of outcomes (eg. requiring high flow oxygenation vs requiring non-invasive ventilation vs requiring invasive mechanical ventilation)?

Minor comments:

1. It is unclear to me how many primary care settings will have access to IL6 measurement.

2. In the introduction the authors should note that non-invasive ventilation/ high flow nasal cannula oxygenation may be injurious and result in patient self induced lung injury.

Reviewer #2: In this manuscript, the authors determined the risk factors for HFNC use in patients with COVID-19 by examining the medical records of patients during the COVID-19 Alpha, Beta, and Delta surges in Japan. They found that among the 19 potential variables, SpO2 <92% (odds ratio [OR] 7.50, 95% confidence interval [CI] 2.806–20.82) and IL-6 (OR 1.021, 95% CI 1.010–1.033) were included in developing the risk score, which was termed IBC-S score. Although it is a well-written manuscript evaluating the severity of COIVD-19, I interpreted that several serious concerns existed.

1) Although they tried to determine the risk factors for needing HFNC support, they evaluated patients who needed oxygenation flow rate ≥10 L/min, high-flow oxygenation, noninvasive ventilation, invasive ventilation, and death. I do not think these patients are representative of patients that need HFNC.

2) They included measurement of IL-6 in their score. I do not think use of HFNC is limited to special hospitals, because it is already used at home. When the patient is in a hospital where they can measure IL-6, I guess they can use HFNC.

6. PLOS authors have the option to publish the peer review history of their article (what does this mean?). If published, this will include your full peer review and any attached files.

Reviewer #1: No

Reviewer #2: **Yes: **Kiyoshi Moriyama

---

## [Author Response · Author response to Decision Letter 0]

7 Oct 2023

5. Review Comments to the Author

Reviewer #1: The authors present their single-center retrospective observational cohort study aiming to identify predictors of respiratory failure. I found this to be a simple and elegant study.

Major comments:

1. The definition of the primary outcome (respiratory failure) covers a wide spectrum of pathology from requiring more than 10L/ min oxygen to requiring mechanical intubation or death. I don't find this to be clinically useful. Is it possible for the authors to look at more meaningful clinical categories of outcomes (eg. requiring high flow oxygenation vs requiring non-invasive ventilation vs requiring invasive mechanical ventilation)?

→ Thank you for your comment. As per your comment, we changed the primary outcome from the critically respiratory illness (composite outcome, including oxygenation flow rate ≥10 L/min, high-flow oxygenation, noninvasive ventilation, invasive ventilation, and death) to require high-flow nasal cannula (HFNC), and redid our statistical analysis.

We revised manuscript and table as follows: 

Line 25:

The primary outcome is the need for HFNC.

Line 30:

The rate of the need for HFNC was 22.9%.

Line 35-36: 

The IBC-S score, an easy-to-use risk score based on parameters available at the time of hospital admission, predicted the need for HFNC in patients with COVID-19. 

Line 66:

The primary outcome is the need for HFNC.

Line 68-69:

We excluded patients who needed oxygenation flow rate ≥10 L/min, needed HFNC and mechanical ventilation, or died on admission day.

Line 116-119:

Of these 152 patients, 4 patients who needed oxygenation flow rate ≥10 L/min, needed HFNC and mechanical ventilation, or died on admission day were excluded; therefore, the final analysis included 148 patients.

Line 121-122:

Fig 1 Study flow diagram

HFNC: high-flow nasal cannula; COVID-19: coronavirus disease 2019

Line 126-127:

Of the 148 patients, 34 patients (22.9%) required HFNC, 9 (6.1%) patients required invasive ventilation, and 13 patients (8.8%) died.

Table 1:

We added the clinical course (HFNC, mechanical ventilation and death).

Line 131-136：

A total of 19 variables that were measured at the time of admission were included in the least absolute shrinkage and selection operator regression, which identified SpO2 and IL-6 as significant predictors of the need for HFNC. In the logistic regression model including these two parameters, both an SpO2 <92% (odds ratio [OR] 7.50, 95% confidence interval [CI] 2.806–20.82; P < 0.001) and IL-6 (per 10 pg/ml increase, OR 1.23, 95% CI 1.010–1.038; P < 0.001) were independent and significant predictors of the need for HFNC and were included in the risk score (Table 2).

Table2: 

Table 2 Association of SpO2 <92% and IL-6 with the risk of the need for HFNC

Line 138-140:

The cut-off IL-6 value of 19.5 pg/mL exhibited equal sensitivity and specificity with the highest positive predictive value of 63.2 pg/mL; therefore, this value was used to create the IL-6-based COVID-19 severity [IBC-S] score to determine the risk of the need for HFNC (Table 3).

Table3:

Table 3 IL-6 based COVID-19 severity score to predict the need for HFNC

Minor comments:

1. It is unclear to me how many primary care settings will have access to IL6 measurement.

→ Thank you for your comment. IL-6 can be measured at common testing facilities, and IL-6 measurement ELISA kits are available; thus, it can be measured in primary care settings as well.

We revised manuscript as follows:

Line 176-178:

IL-6 can be measured at common diagnostic facilities, and there are commercially available IL-6 measurement ELISA kits, enabling measurement in primary care settings.

2. In the introduction the authors should note that non-invasive ventilation/ high flow nasal cannula oxygenation may be injurious and result in patient self induced lung injury.

→ Thank you for your comment. We have added the following sentence:

Line 46-48:

In contrast, HFNC may promote further lung injury (e.g., self-inflicted lung injury) through vigorous breathing efforts.

Reviewer #2: In this manuscript, the authors determined the risk factors for HFNC use in patients with COVID-19 by examining the medical records of patients during the COVID-19 Alpha, Beta, and Delta surges in Japan. They found that among the 19 potential variables, SpO2 <92% (odds ratio [OR] 7.50, 95% confidence interval [CI] 2.806–20.82) and IL-6 (OR 1.021, 95% CI 1.010–1.033) were included in developing the risk score, which was termed IBC-S score. Although it is a well-written manuscript evaluating the severity of COIVD-19, I interpreted that several serious concerns existed.

1) Although they tried to determine the risk factors for needing HFNC support, they evaluated patients who needed oxygenation flow rate ≥10 L/min, high-flow oxygenation, noninvasive ventilation, invasive ventilation, and death. I do not think these patients are representative of patients that need HFNC.

→ Thank you for your comment. As per your comment, we changed the primary outcome from the critically respiratory illness (composite outcome, including oxygenation flow rate ≥10 L/min, high-flow oxygenation, noninvasive ventilation, invasive ventilation, and death) to require high-flow nasal cannula (HFNC), and redid our statistical analysis.

We revised manuscript and table as follows: 

Line 25:

The primary outcome is the need for HFNC.

Line 30:

The rate of the need for HFNC was 22.9%.

Line 35-36: 

The IBC-S score, an easy-to-use risk score based on parameters available at the time of hospital admission, predicted the need for HFNC in patients with COVID-19. 

Line 66:

The primary outcome is the need for HFNC.

Line 68-69:

We excluded patients that needed oxygenation flow rate ≥10 L/min, needed HFNC, needed mechanical ventilation or died on admission day.

Line 116-119:

Of these 152 patients, 4 patients who needed oxygenation flow rate ≥10 L/min, needed HFNC, needed mechanical ventilation or died on admission day were excluded; therefore, the final analysis included 148 patients.

Line 121-122:

Fig 1 Study flow diagram

HFNC, high-flow nasal cannula; COVID-19, coronavirus disease 2019

Line 126-127:

Of the 148 patients, 34 patients (22.9%) required HFNC, 9 (6.1%) patients required invasive ventilation, and 13 patients (8.8%) died.

Table 1:

We added the clinical course (HFNC, mechanical ventilation and death).

Line 131-136：

A total of 19 variables that were measured at the time of admission were included in the least absolute shrinkage and selection operator regression, which identified SpO2 and IL-6 as significant predictors of the need for HFNC. In the logistic regression model including these two parameters, both an SpO2 <92% (odds ratio [OR] 7.50, 95% confidence interval [CI] 2.806–20.82; P < 0.001) and IL-6 (per 10 pg/ml increase, OR 1.23, 95% CI 1.010–1.038; P < 0.001) were independent and significant predictors of the need for HFNC and were included in the risk score (Table 2).

Table2: 

Table 2 Association of SpO2 <92% and IL-6 with the risk of the need for HFNC

Line 138-140:

The cut-off IL-6 value of 19.5 pg/mL exhibited equal sensitivity and specificity with the highest positive predictive value of 63.2 pg/mL; therefore, this value was used to create the IL-6-based COVID-19 severity [IBC-S] score to determine the risk of the need for HFNC (Table 3).

Table3:

Table 3 IL-6 based COVID-19 severity score to predict the need for HFNC

2) They included measurement of IL-6 in their score. I do not think use of HFNC is limited to special hospitals, because it is already used at home. When the patient is in a hospital where they can measure IL-6, I guess they can use HFNC.

→ Thank you for your comment. As mentioned by you, use of HFNC is not limited to special hospitals, because it is already used at home.

Thus, we deleted the sentence.

Line 51 before revise

However, HFNC uses hyperbaric oxygen, which limits the medical facilities that can utilize it.

---

## [Decision Letter · Decision Letter 1]

22 Oct 2023

PONE-D-23-24917R1A risk stratification model for high-flow nasal cannula use in patients with coronavirus disease 2019 in Japan: a single-center retrospective observational cohort studyPLOS ONE

Dear Dr. Kurihara,

Thank you for submitting your manuscript to PLOS ONE. After careful consideration, we feel that it has merit but does not fully meet PLOS ONE’s publication criteria as it currently stands. Therefore, we invite you to submit a revised version of the manuscript that addresses the points raised during the review process.

We look forward to receiving your revised manuscript.

Kind regards,

Gilbert Sterling Octavius

Academic Editor

PLOS ONE

Reviewers' comments:

Reviewer's Responses to Questions

**Comments to the Author**

1. If the authors have adequately addressed your comments raised in a previous round of review and you feel that this manuscript is now acceptable for publication, you may indicate that here to bypass the “Comments to the Author” section, enter your conflict of interest statement in the “Confidential to Editor” section, and submit your "Accept" recommendation.

Reviewer #1: All comments have been addressed

Reviewer #2: (No Response)

2. Is the manuscript technically sound, and do the data support the conclusions?

Reviewer #1: Yes

Reviewer #2: No

3. Has the statistical analysis been performed appropriately and rigorously? 

Reviewer #1: I Don't Know

Reviewer #2: No

4. Have the authors made all data underlying the findings in their manuscript fully available?

Reviewer #1: Yes

Reviewer #2: Yes

5. Is the manuscript presented in an intelligible fashion and written in standard English?

Reviewer #1: Yes

Reviewer #2: Yes

6. Review Comments to the Author

Reviewer #1: The authors have responded to all of the comments. The manuscript makes a helpful contribution to the literature.

Reviewer #2: According to the suggestions by the reviewer, the authors changed their primary outcome from the critically respiratory illness (composite outcome, including oxygenation flow rate ≥10 L/min, high-flow oxygenation, noninvasive ventilation, invasive ventilation, and death) to require high-flow nasal cannula (HFNC), and redid their statistical analysis.

Major comments

1) It looks strange to me that their non-outcome group contains patients that did not require HFNC and 9 patients that required invasive ventilation, and 13 patients that died. How is the disease severity of the non-outcome group?

2) If the introduction of HFNC is the primary outcome, please explain the introducing criteria of HFNC at your institution.

Minor comments

Please explain what "outcome" and "non-outcome" mean in the Tables and Figures.

7. PLOS authors have the option to publish the peer review history of their article (what does this mean?). If published, this will include your full peer review and any attached files.

Reviewer #1: No

Reviewer #2: **Yes: **Kiyoshi Moriyama

---

## [Author Response · Author response to Decision Letter 1]

5 Nov 2023

5. Review Comments to the Author

Reviewer #1: The authors have responded to all of the comments. The manuscript makes a helpful contribution to the literature.

→ Thank you for your valuable comment.

Reviewer #2: According to the suggestions by the reviewer, the authors changed their primary outcome from the critically respiratory illness (composite outcome, including oxygenation flow rate ≥10 L/min, high-flow oxygenation, noninvasive ventilation, invasive ventilation, and death) to require high-flow nasal cannula (HFNC), and redid their statistical analysis.

Major comments

1) It looks strange to me that their non-outcome group contains patients that did not require HFNC and 9 patients that required invasive ventilation, and 13 patients that died. How is the disease severity of the non-outcome group?

→ Thank you for your insightful comment.

Table 1 shows that “non-outcome group (n = 114)” comprises no patients who required HFNC, invasive ventilation, or died, and that “outcome group (n = 34)” consisted of 34 patients who required HNFC for the primary outcome, of which 9 patients required invasive ventilation and 12 patients died.

2) If the introduction of HFNC is the primary outcome, please explain the introducing criteria of HFNC at your institution.

→ Thank you for your valuable comment. Accordingly, we have added the sentence and two references for explaining the criteria for introduction of HFNC at our institution as follows;

Page 4, Line 66-69

HFNC was administered to patients who were unable to maintain an SpO2 level of above 92% and respiratory rate below 25 breaths per minute, even when they received standard oxygen administered through a face mask at the flow rate of 10 L/min or higher [8,9].

8. Xu J, Yang X, Huang C, Zou X, Zhou T, Pan S, et al. A novel risk-stratification models of the high-flow nasal cannula therapy in COVID-19 patients with hypoxemic respiratory failure. Front Med (Lausanne). 2020;7:607821. doi: 10.3389/fmed.2020.607821, PMID 33425951.

9. Roca O, Caralt B, Messika J, Samper M, Sztrymf B, Hernández G, et al. An index combining respiratory rate and oxygenation to predict outcome of nasal high-flow therapy. Am J Respir Crit Care Med. 2019;199(11):1368-76. doi: 10.1164/rccm.201803-0589OC, PMID 30576221.

Minor comments

Please explain what "outcome" and "non-outcome" mean in the Tables and Figures.

→ Thank you for your valuable comment. Accordingly, we have marked "outcome" in Table 1 with one asterisk and “non-outcome” with two asterisks, and described the meaning of each in the footnotes as follows. Additionally, we have added the sentence in Fig 1 caption as follows:

Table1:

*Indicates requirement of HFNC, and **Indicates no requirement of HFNC.

Fig 1 Study flow diagram. Outcome indicates requirement of HFNC, Non-Outcome indicates no requirement of HFNC.

HFNC: high-flow nasal cannula; COVID-19: coronavirus disease 2019

---

## [Decision Letter · Decision Letter 2]

13 Nov 2023

A risk stratification model for high-flow nasal cannula use in patients with coronavirus disease 2019 in Japan: a single-center retrospective observational cohort study

PONE-D-23-24917R2

Dear Dr. Kurihara,

We’re pleased to inform you that your manuscript has been judged scientifically suitable for publication and will be formally accepted for publication once it meets all outstanding technical requirements.

Kind regards,

Gilbert Sterling Octavius

Academic Editor

PLOS ONE

Additional Editor Comments (optional):

Reviewers' comments:

Reviewer's Responses to Questions

**Comments to the Author**

1. If the authors have adequately addressed your comments raised in a previous round of review and you feel that this manuscript is now acceptable for publication, you may indicate that here to bypass the “Comments to the Author” section, enter your conflict of interest statement in the “Confidential to Editor” section, and submit your "Accept" recommendation.

Reviewer #2: All comments have been addressed

2. Is the manuscript technically sound, and do the data support the conclusions?

Reviewer #2: Yes

3. Has the statistical analysis been performed appropriately and rigorously? 

Reviewer #2: Yes

4. Have the authors made all data underlying the findings in their manuscript fully available?

Reviewer #2: Yes

5. Is the manuscript presented in an intelligible fashion and written in standard English?

Reviewer #2: Yes

6. Review Comments to the Author

Reviewer #2: (No Response)

7. PLOS authors have the option to publish the peer review history of their article (what does this mean?). If published, this will include your full peer review and any attached files.

Reviewer #2: No

---

## [Editor Report · Acceptance letter]

16 Nov 2023

PONE-D-23-24917R2 

A risk stratification model for high-flow nasal cannula use in patients with coronavirus disease 2019 in Japan: a single-center retrospective observational cohort study 

Dear Dr. Kurihara:

I'm pleased to inform you that your manuscript has been deemed suitable for publication in PLOS ONE. Congratulations! Your manuscript is now with our production department. 

Kind regards, 

on behalf of

Dr. Gilbert Sterling Octavius 

Academic Editor

PLOS ONE